# Characterization and Preliminary Application of a Novel Lipoxygenase from *Enterovibrio norvegicus*

**DOI:** 10.3390/foods11182864

**Published:** 2022-09-15

**Authors:** Bingjie Zhang, Meirong Chen, Bingjie Xia, Zhaoxin Lu, Kuan Shiong Khoo, Pau Loke Show, Fengxia Lu

**Affiliations:** 1College of Food Science and Technology, Nanjing Agricultural University, Nanjing 210095, China; 2Department of Chemical Engineering and Materials Science, Yuan Ze University, Taoyuan 32003, Taiwan; 3Faculty of Science and Engineering, University of Nottingham Malaysia, Jalan Broga, Semenyih 43500, Malaysia

**Keywords:** lipoxygenase, *Enterovibrio norvegicus*, characterization, application

## Abstract

Lipoxygenases have proven to be a potential biocatalyst for various industrial applications. However, low catalytic activity, low thermostability, and narrow range of pH stability largely limit its application. Here, a lipoxygenase (LOX) gene from *Enterovibrio norvegicus* DSM 15893 (EnLOX) was cloned and expressed in *Escherichia coli* BL21 (DE3). EnLOX showed the catalytic activity of 40.34 U mg^−1^ at 50 °C, pH 8.0. Notably, the enzyme showed superior thermostability, and wide pH range stability. EnLOX remained above 50% of its initial activity after heat treatment below 50 °C for 6 h, and its melting point temperature reached 78.7 °C. More than 70% of its activity was maintained after incubation at pH 5.0–9.5 and 4 °C for 10 h. In addition, EnLOX exhibited high substrate specificity towards linoleic acid, and its kinetic parameters of V_max_, K_m_, and K_cat_ values were 12.42 mmol min^−1^ mg^−1^, 3.49 μmol L^−1^, and 16.86 s^−1^, respectively. LC-MS/MS analysis indicated that EnLOX can be classified as 13-LOX, due to its ability to catalyze C_18_ polyunsaturated fatty acid to form 13-hydroxy fatty acid. Additionally, EnLOX could improve the farinograph characteristics and rheological properties of wheat dough. These results reveal the potential applications of EnLOX in the food industry.

## 1. Introduction

Lipoxygenases (LOXs) are a family of non-heme iron dioxygenases, which catalyze the regiospecific and stereospecific insertion of molecular oxygen in polyunsaturated fatty acids (PUFAs), containing 1-*Z*, 4-*Z*-pentadiene moieties into hydroperoxides fatty acid with (*Z*), (*E*)-diene conjugations [1,2,3]. LOXs have a broad potential for application in food, chemical, and pharmaceutical industries, of which the application prospects in dough products are due to the reinforcement effect by enhancing the gluten network [4,5]. LOX is extracted mainly from soybean powder [6,7,8,9], which contains multiple enzymes, and may reduce the catalyzing effects of LOX and produce unpleasant odor after the catalytic process, thus, limiting the application of LOXs in flour products [7,10,11]. Hence, the hunt for LOXs with desired properties has gained widespread attention for its application prospects in dough products.

LOXs are widely found in animals, plants, and microorganisms, but only microbial-LOXs are of industrial significance due to its high yield, low cost, and simple preparation process [12,13]. To date, many LOXs from microorganisms have been characterized [14,15], especially LOXs from *Anabaena* [16], *Burkholderia thailandensis* [15], *Pseudomonas aeruginosa* [14], and *Myxococcus xanthus* [1]. However, these LOXs cannot meet all the requirements of high catalytic activity, good thermostability, and wide pH-stability simultaneously. Directed evolution, rational design, and semi-rational design were used to improve biochemical characteristics of LOX [17,18,19,20]. Consensus design method was used to improve the thermostability of LOX from *Anabaena* [21]; the specific activity of *P. aeruginosa* LOX was improved using the fused self-assembling amphipathic peptides [19]. Although researchers have made a lot of efforts to improve the properties of LOXs by protein engineering [17,19,21], LOX, with the comprehensive properties, has not yet been obtained. Meanwhile, these methods are dependent on the analysis of crystal structures or the establishment of a high-throughput screening library, and it is hard to obtain mutant enzymes with superior properties in a short period of time [20]. Therefore, microbial-LOXs have been reported to not be suitable for industrial application, which calls for high yield, high purity, acid and alkali resistance, good thermal stability, and high catalytic activity [7,21]. Considering the demand of LOXs with satisfactory properties, it is reasonable to gain new types of high-efficiency, thermo-stable, and wide pH-range stable LOXs by excavating the genetic reservoir in nature [20,22].

In this study, a novel LOX gene from *E. norvegicus* DSM 15893 was analyzed using bioinformatics technology, then cloned and expressed in *Escherichia coli* BL21 (DE3). The properties of the recombinant *E. norvegicus* LOX (EnLOX) have been characterized. Moreover, the influence of EnLOX on the farinograph characteristics and rheological properties of dough were evaluated.

## 2. Materials and Methods

### 2.1. Strains, Plasmids, and Reagents and Study Design

The LOX coding gene from *E. norvegicus* DSM 15893 (accession number WP_074927588) was codon-optimized and cloned to the plasmid pET-28a (+) with *Nde* I and *Xho* I, as the inserted restriction sites by Genewiz. *E. coli* strains DH5α (Novozymes, Wilmington, USA) and BL21 (DE3) (Novozymes, Wilmington, USA) were preserved in our laboratory. Kanamycin and isopropyl-β-_D_-thiogalactopyranoside (IPTG) were purchased from Solarbio (Beijing, China). PUFA standards, including linoleic acid (LA), α-linolenic acid (ALA), and γ-linolenic acid (GLA), were purchased from Sigma (Steinheim, Germany). Hydroxy fatty acid (HFA) standards, including 13-hydroxyoctadecaenoic acid (13-HODE), 13(S)-hydroxyoctadeca trienoic acid (13S-HOTrE), were purchased from Cayman Chemical (Ann Arbor, MI, USA). All other chemicals were analytical grade. The study design was shown in Figure 1.

### 2.2. Phylogenetic Analysis and Structural Modeling of EnLOX

SWISS-MODEL was used to build the three dimensional (3D) model [23]. Pymol was used to analyze the 3D structure of EnLOX [24]. ProtParam tool of Expasy was used to predicted the molecular mass of EnLOX [25]. The nucleotide and amino acid sequences of EnLOX were subjected to homology sequence similarity retrieval and analysis on the NCBI, and then the percent identities among the nucleotide and amino acid sequences were calculated by MEGA 5.05.

### 2.3. Expression and Purification of EnLOX

*E. coli* BL21 (DE3)/pET28a-EnLOX combination was inoculated into the Luria Broth medium (LB) containing 100 μg mL^−1^ kanamycin at 37 °C and agitation rate of 180 rpm until the optical density at 600 nm reached 0.6–0.8. The final concentration of 100 μg mL^−1^ IPTG was added to induce the protein expression at 16 °C for 16 h. The cells were collected by centrifugation (8000× *g*, 5 min), and resuspended in Tris-HCl buffer (0.05 mol L^−1^ Tris-HCl, 0.3 mol L^−1^ NaCl, pH 7.5). After cells were completely broken by D-3L High Pressure Homogenizer (PhD Technology LLC, Minnesota, USA), cell debris were removed by centrifugation (4 °C, 10000× *g*, 30 min). The supernatant loaded onto Ni-NTA affinity column for purification. The purity and molecular mass of EnLOX were checked by sodium dodecyl sulfate polyacrylamide gel electrophoresis (SDS-PAGE) [1,21].

### 2.4. Determination of EnLOX Activity and Protein Concentration

EnLOX activity was assayed in 20 mmol L^−1^ Tris-HCl buffer (pH 8.0) at 50 °C by monitoring the increase in absorbance at 234 nm. The final concentration of 1.73 mmol L^−1^ PUFAs was added to a 3 mL reaction system. One unit of activity was defined as the amount of enzyme required to synthesize 1 mmol hydroperoxide per min with an extinction coefficient of 25,000 L mol^−1^ cm^−1^ [1,20,21,26]. The protein concentration was measured by Bradford method [27,28].

### 2.5. Kinetic Parameters of EnLOX

Substrate specificity of EnLOX was determined with different final concentration of PUFAs from 10 to 300 mmol L^−1^. All measurements were carried out at 50 °C in 20 mmol L^−1^ Tris-HCl buffer (pH 8.0). The kinetic parameters of *K_m_*, *V_max_*, and *K_cat_* were calculated by nonlinear regression fitting of the experimental data to Michaelis–Menten equation using GraphPad software. Michaelis–Menten equation is shown as follows [29,30]:(1)$$v_{0}=\frac{V_{max}\;\left[S\right]}{K_{m}+\left[S\right]}$$
where *v_0_*, *V_max_*, [*S*], and *K_m_* represent the initial rate, the maximum velocity of reaction, the substrate concentration, and the Michaelis constant, respectively.

### 2.6. Product Detection of EnLOX

Product specificity of EnLOX was determined by incubating PUFAs, EnLOX, and Tris-HCl buffer (pH 8.0) at 50 °C for 10 min. Hydroperoxides formed were reduced to their corresponding hydroxides with addition of 10 mmol L^−1^ cysteine. The reaction mixtures were extracted according to the method of *HUANG* et al. [31]. The products were determined by liquid chromatography-tandem mass spectrometry (LC-MS/MS), with an UPLC column (2.1 × 100 mm ACQUITY UPLC BEH C18 column containing 1.7 μm particles). Buffer A consisted of 0.1% formic acid in water; buffer B consisted of 0.1% formic acid in acetonitrile. The gradient was 20% Buffer B for 0.5 min, 20–98% Buffer B for 10 min, and 98% Buffer B for 4 min, with a flow rate of 0.4 mL min^−1^. Mass spectrometry was performed using electrospray source in negative ion mode with MS acquisition mode, with a selected mass range of 100–1000 m/z. The lock mass option was enabled using leucine-enkephalin (m/z 556.2771) for recalibration. The ionization parameters were the following: capillary voltage was 2.5 kV, sample cone was 40 V, source temperature was 120 °C, and desolvation gas temperature was 800 °C. Data acquisition and processing were performed using Masslynx 4.1.

### 2.7. Effect of pH and Temperature on EnLOX Activity and Stability

The effect of pH on EnLOX activity was measured from pH 3.0 to pH 9.5 using citrate sodium citrate buffer (pH 3.0–6.0), phosphate buffer solution (pH 6.0–7.0), Tris-HCl buffer (pH 7.0–9.0), and borax-sodium hydroxide buffer (pH 9.0–9.5) at 50 °C [32,33]. The highest activity of EnLOX as 100% to calculate the relative activities of varied pH value. To determine the pH stability, we measured the residual activity after incubating the enzyme in buffers with different pH at 4 °C for 10 h [3].

The enzyme activity was measured at different temperatures (20–80 °C) to determine the optimum temperature of EnLOX. The maximum activity was set as 100%; the relative activities at different temperatures were plotted to obtain the optimum temperature of EnLOX [34,35]. Thermostability was analyzed by measuring the residual enzyme activity after incubation at different temperatures (20–70 °C) for different times (0–12 h) before a 5 min ice bath [3,21,35]. Enzyme activity without heat treatment was defined as 100% for comparisons.

### 2.8. Thermal Shift Assay of EnLOX

Protein Thermal Shift was performed using Sypro Orange dye from Sigma-Aldrich [36]. The thermal shift assay was conducted in Light Cycler 480 Real Time Detection System, designed for PCR, for melting point temperature analysis. An amount of 1 μL of 1 ng μL^−1^ of EnLOX, 2 μL of 10X Sypro orange, and 16 μL of 40 mmol L^−1^ Tris-HCl (pH 8.0) were added to the wells of the 96-well Cycler PCR plate. The plate was heated from 25 to 95 °C with a heating rate of 0.5 °C min^−1^. The melting point temperature (*T_m_*) tested is the results of multiple independent experiments, each in triplicate and reported as mean ± standard deviation (SD).

### 2.9. Effect of Metal Ions and Solvents on EnLOX Activity

To investigate the effect of metal ions and solvents on EnLOX activity, different metal ions (CaCl_2_, ZnCl_2_, KCl, MgCl_2_, FeCl_3_, FeCl_2_, MnCl_2_, NaCl, LiCl, (NH_4_)_2_SO_4_, CuSO_4_) with a final concentration of 0.1 mmol L^−1^ and various solvents (phenylmethanesulfonyl fluoride (PMSF), dimethyl sulfoxide (DMSO), sodium dodecyl sulfate (SDS), ethylenediaminetetraacetic acid (EDTA), IPTG, β-mercaptoethanol (β-ME), and Tween 20) with a final concentration of 1.0 mmol L^−1^ were added to the reaction system. The residual activity was measured after 4 °C of incubation for 30 min [1,26,37,38]. The activity of EnLOX without treatment by any metal ions or solvents was defined as 100%.

### 2.10. Farinographical Properties

Farinographical properties of wheat flour blends were examined using a Brabender farinograph-E according to AACC 54–21 method [39]. Wheat flour was put into the mixing bowl of the farinograph after being weighed according to its water absorption, then the sample solutions were added, and then the dough was kneaded for 20 min at 30 °C [40]. Water absorption (WA), dough development time (DDT), stability time (ST), and softening degree (SD) were determined [11]. Sample solution was distilled water contained 50 μg g^−1^ potassium bromate (KBrO_3_), 0 (blank control), 5, 10, 20, 30, 40, 50 IU g^−1^ EnLOX, respectively [4,7]. Mean values of three measurements were reported.

### 2.11. Dynamic Rheological Properties

Rheological properties were measured by using a Physica MCR301 rheometer with a parallel plate (40 mm diameter, 2.5 mm gap), according to the reported research with some modifications [4,41]. The wheat dough was prepared by Brabender farinograph-E, as described before. The dough was taken out immediately and stored at 25 °C for 1 h with plastic film sealing. The rim of the dough was coated with paraffin oil to prevent dryness of the sample. The dough was relaxed for another 5 min to relax the residual stress during sample loading. The tests were measured at 25 °C. The linear viscoelastic zone was tested by stress sweeps (0.01–30%) at 1.0 Hz frequency. Frequency scan tests were performed from 0.1 to 20.0 Hz at constant strain amplitude (30% strain). Data were reported as the means of three measurements.

### 2.12. Statistical Analysis

All the tests were performed in triplicate, and data were expressed as mean ± SD. SPSS 21.0 were used to analyze the differences between the variables. Statistical differences in correlation analysis and linear regression analysis were determined by one-way analysis of variance (ANOVA), and *Duncan*’s test and the significance level of *p* < 0.05 were used [42,43].

## 3. Results and Discussion

### 3.1. Bioinformatic Analysis of EnLOX

The EnLOX sequence resides in an open reading frame encoding 726 amino acids, with the calculated molecular mass of 81.45 kDa. The nucleic acid sequences of original and codon-optimized EnLOX were illustrated in Appendix A. A phylogenetic tree of LOXs, which were reported and functionally identified, was constructed (Figure 2). Analysis of amino acid sequences in Figure 2a revealed that EnLOX had the maximum identity with *E. norvegicus* LOX (WP_017011269). Nevertheless, no characterization experiments of LOX from *E. norvegicus* have been reported so far. The nucleotide sequences showed significant differences between EnLOX and LOXs from other sources (Figure 2b). The differences for LOX genes between nucleotide and amino acid levels indicated the distinction among the organisms. As aligned with amino acid sequences of the characterized linoleate 13-lipoxygenases from soybean, *P. aeruginosa*, *Calothrix*, and *Rivularia* (Appendix A), the catalytic key residues of EnLOX (His404-His409-His601-Asn605-Ile726) are completely conserved.

In order to identify the structural differences with other LOXs, a 3D structural model of EnLOX was predicted. The 3D structure of EnLOX was predicted by Swiss-Model using *P. aeruginosa* LOX (PDB ID: 4rpe) structure as a template, which revealed that EnLOX was composed of 44.0% α-helix, 8.4% strands, and 47.6% random coil (Figure 3a). The proportion of α-helix in an EnLOX structure is similar to *P. aeruginosa* LOX (PDB ID: 4rpe) and *Cyanothece* LOX (PDB ID: 5med). The results also suggested that EnLOX had the lowest (21.76%) and the highest (37.47%) identity with *Gaeumannomyces graminis* LOX (PDB ID: 5fx8) and *P. aeruginosa* LOX, which indicated that it was a novel LOX. Soy flour is used in flour and dough products due to its abundant of soybean LOXs [44]. Soybean LOX-3 (PDB ID: 1rrl) shared the highest structural homology with EnLOX among all of the soybean LOXs. The comparison of the 3D structures of soybean LOX-3 and EnLOX is shown in Figure 3b. EnLOX has only one domain instead of two, as in soybean LOX-3. The N-terminal domain in the soybean LOX-3 is β-folds, which is substituted by α-helices in EnLOX. Increasing α-helix ratio could improve the stability of LOXs [20]. The α-helix ratio of EnLOX is higher than that of most LOXs, such as human arachidonate 12-LOX (α-helix ratio: 38.3%, PDB ID: 3d3l), porcine leukocyte 12S-LOX (39.8%, 3rde), and soybean LOX-1 (31.8%, 4wfo), which may contribute to thermostability of the enzyme.

### 3.2. Expression and Purification of EnLOX

The purified EnLOX showed a single band with a molecular mass of approximately 80 kDa in SDS-PAGE (Figure 4), which was consistent with the predicted value 81.45 kDa. The biochemical properties of EnLOX and LOXs from various sources are shown in Appendix A. The molecular mass of the enzyme was similar to those of *B. thailandensis* (75 kDa) [15] and *M. xanthus* (80 kDa) [1]. The specific activity of purified EnLOX was 40.34 U mg^−1^, which was higher than that of most other sources of LOXs, such as LOXs from *Anabaena* (10.4 U mg^−1^) [21], *B. thailandensis* (26.4 U mg^−1^) [15], and *P. aeruginosa* (28.5 U mg^−1^) [17] but lower than that of *Calothrix* (73.1 U mg^−1^) [26], *Rivularia* (68.8 U mg^−1^) [26], and *A. aegerita* LOX (51.34 U mg^−1^) [35]. In addition, the total yield of the purification for EnLOX was 55.3%, with a purification fold of 31.29.

### 3.3. Kinetic Properties of EnLOX

Kinetic parameters of EnLOX were determined, which showed a discrepancy in the kinetic properties for various PUFAs (Table 1). The *V_max_*, *K_m_*, *K_cat_*, and *K_cat_/K_m_* values of LA were 12.42 mmol min^−1^ mg^−1^, 3.49 μmol L^−1^, 16.86 s^−1^, and 4.83 L μmol^−1^ s^−1^, respectively. The *V_max_* value of LA was the highest among the substrates, indicating that EnLOX had the strongest catalytic rate to LA. This is consistent with the LOXs from *basidiomycete fungus* [2] and *M. xanthus* [45]. The *K_m_* value of LA was the lowest among PUFAs, which exhibited the highest affinity. EnLOX showed a lower *K_m_* than most of the LOXs, such as LOXs from *Calothrix* (6.7 μmol L^−1^) [26], *Rivularia* (7.5 μ mmol L^−1^) [26], *T. bouteillei* (16.5 μmol L^−1^) [26], and *Anabaena* (30.3 μmol L^−1^) [21] and similar to *Nicotiana benthamiana* LOX (3.9 μmol L^−1^) [31]. The *K_cat_/K_m_* value for LA of EnLOX were significantly higher than soybean LOX (0.17 L μmol^−1^ s^−1^) [46], *M. xanthus* LOX (0.02 L μmol^−1^ s^−1^) [45], *Anabaena* (0.54 L μmol^−1^ s^−1^) [21], and *Pleurotus ostreatus* LOX (0.2 L μmol^−1^ s^−1^) [47] but lower than of LOXs from *Calothrix* (13.85 L μmol^−1^ s^−1^) [26] and *Rivularia* (10.79 L μmol^−1^ s^−1^) [26]. The results revealed that LA is the natural substrate for EnLOX.

### 3.4. Products Analysis of EnLOX

Products of EnLOX catalyzed PUFAs were analyzed by LC-MS/MS (Table 2). The main products of LA and ALA formation catalyzed by EnLOX were 13-HODE and 13S-HOTrE, respectively. The products of GLA catalyzed by EnLOX were both 13-HODE and 13S-HOTrE. Most LOXs from plants, such as pear LOX [42], rice seed LOX [48] and Cucumis sativus LOX [43], catalyzed PUFAs, leading to the fission of the C_12_-C_13_ double bond, and oxygenation at position C_13_ to form 13-HFAs [2,43,48,49]. The results suggested that EnLOX can be classified as a 13-LOX.

### 3.5. Effects of Temperature and pH on the Activity and Stability of EnLOX

The activity of EnLOX was measured at different temperatures, as shown in Figure 5a. The optimum temperature of most LOXs varied from 30–40 °C [1,16,35,48,49], however, EnLOX showed the highest catalytic activity at 50 °C. At present, one of the disadvantages for the industrial application of LOX to overcome is the poor thermostability [19]. Noteworthily, EnLOX exhibited high thermostability; more than 50% of the original activity remained after being stored at 20 °C–50 °C for 6 h (Figure 5c). The half-life (T_1/2_) of EnLOX at 50 °C was about 6 h, which was much longer than that of LOXs from soybean (T_1/2_ = 30 min at 45 °C) [50], *M. xanthus* (7.1 min, 50 °C) [1], *P. aeruginosa* (10 min, 50 °C) [19], and *Anabaena* (3.8 min, 50 °C) [21]. The thermal shift assay of EnLOX was completed to further verify the thermostability of the enzyme (Appendix A). The melting point temperature (*T_m_*) value of EnLOX was 78.7 °C, much higher than that of human 5-LOX (36.5 °C) [36] and human 15-LOX (50.0 °C) [51], which revealed the superior thermostability of EnLOX [52]. These results suggested EnLOX is thermotolerant and has potential value in industrial applications.

The activity of EnLOX was assayed at different pHs, as shown in Figure 4b. EnLOX showed activity in a wide range of pH 3.5–9.5 and had a maximum activity at pH 8.0. This result was consistent with the most reported optimum pH of LOXs, which are weakly alkaline [16,35,48,49]. EnLOX displayed excellent stability in assayed pH range (4.0–9.5); it could retain more than 50% relative activity after being incubated at 4 °C for 10 h, and it could retain above 70% relative activity at a broad pH range (5.0–9.5). Surprisingly, EnLOX remained above 90% relative activity at pH 7.5–8.0 after being stored at 4 °C for 10 h (Figure 5d). Excellent stability over a broad pH range may be due to the reversible denaturation of the enzyme, so that there is weak effect on the EnLOX activity on incubation at different pH [53]. Nevertheless, *M. xanthus* LOX is only stable in the range of pH 3.0–6.0 [1]. LOXs from soybean [46], *B. thailandensis* [15], and *Cyanothece* [54] are only stable under alkaline conditions. Therefore, EnLOX could be well-suited for industrial production, as it was stable in weak acidic and alkaline environment.

### 3.6. Effect of Metal Ions and Solvents on EnLOX Activity

As shown in Table 3, the activity of EnLOX varies with the addition of different metal ions and solvents. Ca^2+^, Mn^2+^, Fe^3+^, and DMSO enhanced EnLOX activity by 2.37, 2.12, 1.89, and 2.13 times, respectively. The promotion of enzyme activity by metal ions possibly related to the N-terminal PLAT domain, cofactors, and coenzyme of EnLOX [55,56]. These results were similar to *M. xanthus* LOX [1] and *Cyanobacterium Cyanothece* LOX [55]. Zn^2+^, Li^+^, and K^+^ had no effects on the enzyme activity. The activating effect on EnLOX of DMSO may be due to DMSO changing some secondary bonds in the enzyme and influencing the flexibility of the activity site [38]. The enzyme activity was completely inhibited by EDTA, SDS, and β-ME. EDTA could chelate iron ion in LOX, thus EnLOX was proved to be a metal enzyme [57]. Some solvents could affect iron ion or change the conformation of LOX, thus, showing the inhibitory effect of the enzyme activity [58]. The activation function of Ca^2+^, Mn^2+^, Fe^3+^, and DMSO provide new methods for the industrial application of EnLOX to optimize the fermentation conditions.

### 3.7. Effect of EnLOX on Farinographical Properties

The effects of EnLOX on the farinographical properties of wheat flour are shown in Table 4. Compared with the blank group, water absorption, dough development time, and stability time of flours fortified with 5–30 IU g^−1^ EnLOX increased, softening degree decreased. Water absorption, dough development time, and stability time of flours fortified with 20 IU g^−1^ EnLOX increased by 3.89%, 0.50 min, and 1.14 min, respectively; softening degree was reduced by 39 FU. Fatty acid hydroperoxides generated by EnLOX in the catalytic process can induce protein polymerization [5], and, at the same time, oxidize the sulfhydryl group (-SH) of gluten to the disulfide bond (-S-S-) to the enhanced gluten network [59], which promotes the stability of the wheat dough, manifested as dough development time and stability time prolongation, and decreases the softening degree value [11]. The indicators of flour, which added KBrO_3_, had no significant changes apart from the softening degree decreasing significantly. EnLOX exhibited favorable farinographical properties of wheat flour and could be added in flour products to improve its quality.

### 3.8. Effect of EnLOX on Rheological Properties

Rheological characteristics of dough treated with different concentrations of EnLOX were evaluated (Figure 6). Dough samples added with EnLOX showed a significant increase in elastic value (*G′*) (Figure 6a) and viscous (*G″*) value (Figure 6b); both of the two values elevated with the increase in EnLOX concentration, revealing the good elasticity and extensibility of dough [4]. The degree of polymerization in dough can be described by tan δ (*G″*/*G′*) [60]. With the increase in the frequency, the tan δ value was always lower than 1.0 (Figure 6c), which suggested that elastic properties were more prominent, and the dough exhibits solid phase properties. Dough treated with 10–50 IU g^−1^ EnLOX exhibited higher elastic and viscous characteristics than that of the blank group, indicating that EnLOX could enhance the cross-linking of gluten [60]. EnLOX has been proven to have the reinforcement ability in wheat dough, similar to previously reported *Anabaena* LOX [4] and soybean LOX [46], indicating that the enzyme could be a potential enzymatic preparation [7].

## 4. Conclusions

To overcome the low enzyme activity, poor thermal stability, and narrow pH range stability of existing LOX enzymes, a novel LOX from *Enterobacter norvegicus* was expressed and characterized in this study. EnLOX only showed 37.47% identity with *P. aeruginosa* LOX, and the enzyme exhibited superior properties; good thermostability at 50 °C; outstanding pH stability at pH 5.0–9.5; and high activity of LA, achieving 40.34 U mg^−1^. In addition, the application of EnLOX to wheat dough further improved the farinograph and rheological properties. Our research indicated EnLOX is a potential enzyme agent for food industrial applications.

## Figures and Tables

**Figure 1 foods-11-02864-f001:**
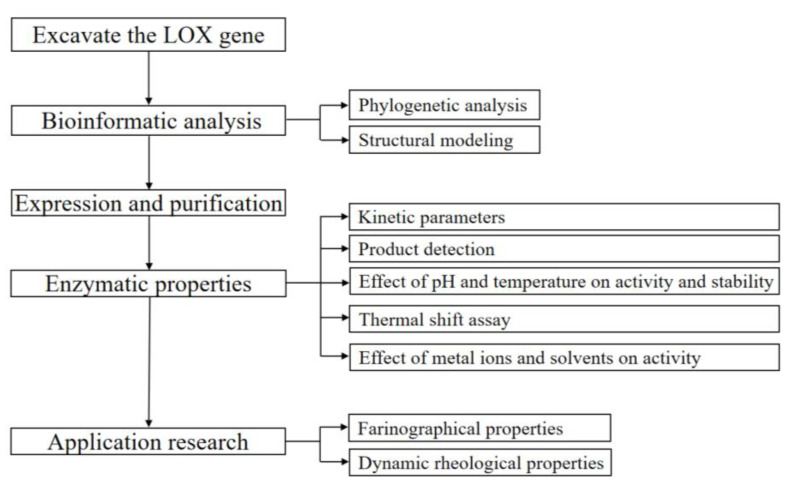
Flow chart of methodology.

**Figure 2 foods-11-02864-f002:**
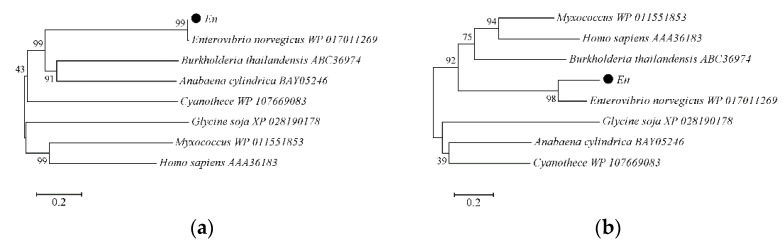
(**a**) Phylogenetic tree of amino acid sequences of EnLOX. (**b**) Phylogenetic tree of nucleotide sequences of EnLOX.

**Figure 3 foods-11-02864-f003:**
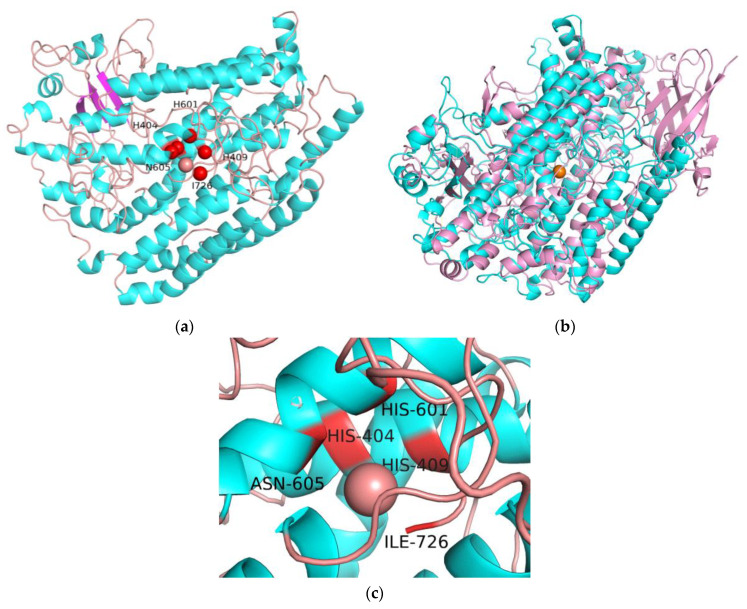
Molecular model depicting predicted structure of EnLOX and molecular model of soybean LOX-3. (**a**) The model structure of EnLOX. The α-helix (blue), strands (pink), and random coil (orange) of EnLOX. (**b**) Superposition of a soybean LOX-3 and an EnLOX. Soybean LOX-3 is shown in pink and EnLOX is shown in blue. (**c**) The catalytic key residues of EnLOX (His404-His409-His601-Asn605-Ile726). The orange sphere represents the iron ion in LOXs.

**Figure 4 foods-11-02864-f004:**
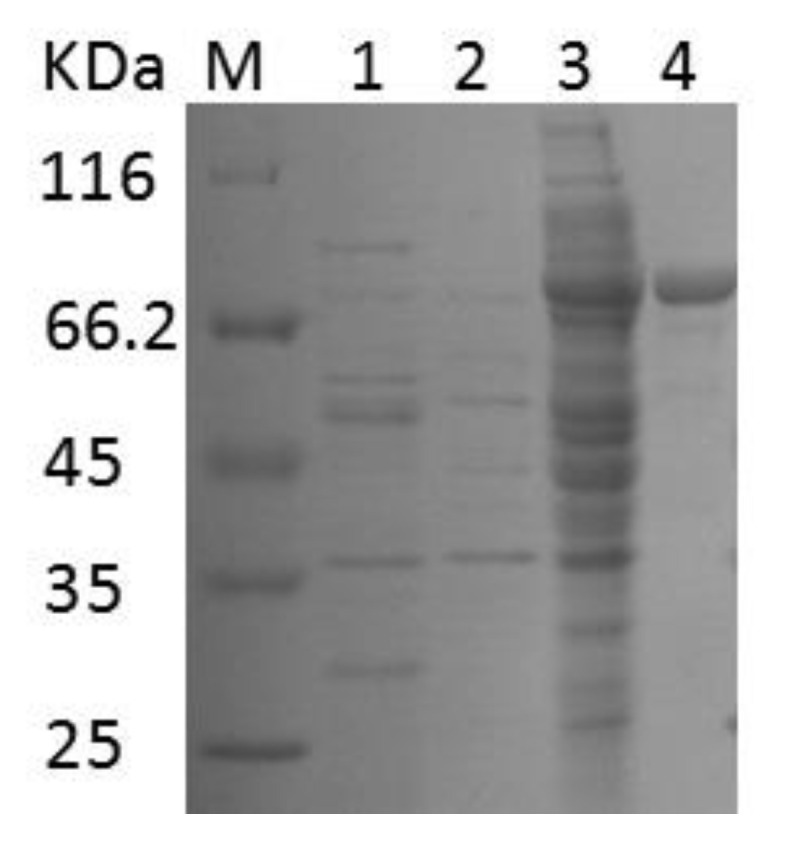
SDS-PAGE analysis of EnLOX expression and purification. M, protein molecular weight standard; Lane 1, *E. coli*/pET28a cell extracts; Lane 2, *E. coli*/pET28a-EnLOX of no-IPTG induced; Lane 3, *E. coli*/pET28a-EnLOX induced by IPTG for 16 h; Lane 4, EnLOX purified by Ni-NTA.

**Figure 5 foods-11-02864-f005:**
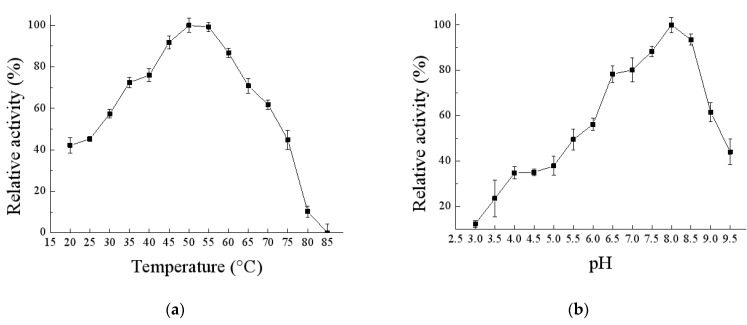
Effect of temperature and pH on the activity and stability of EnLOX. EnLOX activity was plotted as the relative activity (%), with the highest activity as 100%. Data are shown as means ± SD of three independent experiments. (**a**) The optimum temperature of EnLOX was determined by measuring the activity in 50 mmol L^−1^ Tris-HCl (pH 8.0) at 20–85 °C. (**b**) The optimum pH of EnLOX was determined by measuring the activity in pH 3.0–9.5 at 50 °C. (**c**) Thermostability of EnLOX. Enzyme samples were incubated at different temperatures (20–70 °C) for different times (0–12 h) before a 5 min ice bath; the residual activities were measured in 50 mmol L^−1^ Tris-HCl (pH 8.0) at 50 °C. (**d**) The pH stability of EnLOX. Enzyme samples were incubated at each pH at 4 °C for 10 h, the residual activities were measured in 50 mmol L^−1^ Tris-HCl (pH 8.0) at 50 °C.

**Figure 6 foods-11-02864-f006:**
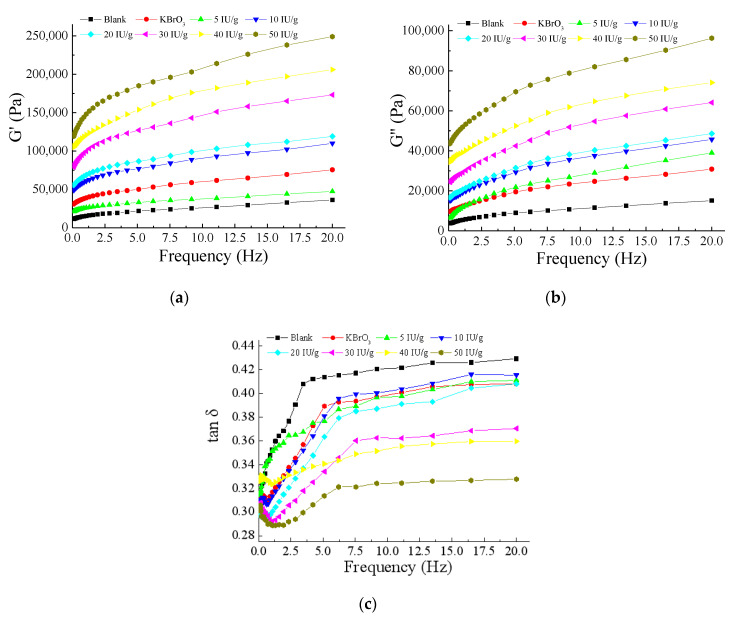
The effect of EnLOX on dough rheology. (**a**) The influence of EnLOX on the elastic modulus (*G′*) of the dough; (**b**) The influence of EnLOX on the viscosity modulus (*G″*) of the dough; (**c**) The influence of EnLOX on the tan δ of the dough.

**Table 1 foods-11-02864-t001:** Kinetic parameters of EnLOX.

Substrate	*V_max_* (mmol min^−1^ mg^−1^)	*K_m_* (μmol L^−1^)	*K_cat_* (s^−1^)	*K_cat_/K_m_* (L μmol^−1^ s^−1^)
LA	12.42	3.49	16.86	4.83
ALA	4.24	2.32	5.76	2.48
GLA	6.23	8.79	8.45	0.96

LA is linoleic acid, ALA is α-linolenic acid, and GLA is γ-linolenic acid.

**Table 2 foods-11-02864-t002:** Analysis of products of PUFAs catalyzed by EnLOX.

Substrate	Class	Product	MS/MS Fragments [M-H]^−^	Position of Oxygenation
LA	ω-6	13-HODE	295.2273	n-6
ALA	ω-6	13S-HOTrE	293.2117	n-6
GLA	ω-6	13S-HOTrE, 13-HODE	295.2273, 293.2117	n-6

LA is linoleic acid, ALA is α-linolenic acid, GLA is γ-linolenic acid, 13-HODE is 13-hydroxyoctadecaenoic acid, and 13S-HOTrE is 13(S)-hydroxyoctadeca trienoic acid.

**Table 3 foods-11-02864-t003:** Effect of metal ions and solvents on EnLOX activity. EnLOX activity was plotted as the relative activity (%), with the highest activity as 100%.

Metal Ion and Solvent	Relative Activity (%)
Control	100
Ca^2+^	236.8 ± 2.1
Mn^2+^	212.22 ± 6.0
Fe^3+^	189.47 ± 3.8
Zn^2+^	101.24 ± 2.7
Li^+^	98.44 ± 11.0
K^+^	98.11 ± 5.0
Mg^2+^	92.59 ± 5.2
Cu^2+^	84.53 ± 3.5
NH_4_^+^	82.21 ± 1.4
Na^+^	76.85 ± 16.4
Fe^2+^	55.66 ± 9.4
DMSO	213.41 ± 6.9
IPTG	69.15 ± 0.1
PMSF	60.88 ± 4.1
Tween 20	15.05 ± 7.1
EDTA	0
SDS	0
β-ME	0

Data are shown as means ± SD of three independent experiments.

**Table 4 foods-11-02864-t004:** Effect of EnLOX enzyme on farinograph quality of the wheat dough.

Sample	WA (%)	DDT (min)	ST (min)	SD (FU)
Blank	53.04 ± 1.27 ^e^	1.20 ± 0.05 ^d^	2.23 ± 0.12 ^def^	103.33 ± 3.06 ^c^
50 μg g^−1^ KBrO_3_	53.80 ± 0.26 ^de^	1.30 ± 0.05 ^cd^	2.37 ± 0.06 ^cde^	74.33 ± 0.58 ^d^
5 IU g^−1^ EnLOX	55.17 ± 0.15 ^bc^	1.40 ± 0.10 ^c^	2.53 ± 0.06 ^bcd^	77.67 ± 2.08 ^d^
10 IU g^−1^ EnLOX	55.53 ± 0.23 ^b^	1.53 ± 0.05 ^b^	2.60 ± 0.05 ^bc^	75.67 ± 0.58 ^d^
20 IU g^−1^ EnLOX	56.93 ± 0.85 ^a^	1.70 ± 0.10 ^a^	3.37 ± 0.47 ^a^	64.33 ± 2.08 ^e^
30 IU g^−1^ EnLOX	55.93 ± 0.12 ^ab^	1.20 ± 0.05 ^d^	2.73 ± 0.06 ^b^	75.00 ± 1.00 ^d^
40 IU g^−1^ EnLOX	54.43 ± 0.6 ^cd^	1.03 ± 0.05 ^e^	2.17 ± 0.05 ^ef^	116.67 ± 3.79 ^b^
50 IU g^−1^ EnLOX	52.93 ± 0.21 ^e^	0.83 ± 0.17 ^f^	1.99 ± 0.12 ^f^	126.67 ± 1.53 ^a^

Data are shown as means ± SD of three independent experiments. Means with different lowercase letters indicate significant differences between groups (*p* < 0.05). WA is water absorption, DDT is dough development time, ST is stability time, and SD is softening degree.

## Data Availability

Data are contained within the article and available upon reasonable request from the corresponding author.

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
