# Peer review of "Characterization and Preliminary Application of a Novel Lipoxygenase from Enterovibrio norvegicus"

_foods, 2022, doi:10.3390/foods11182864_

Round 1
Reviewer 1 Report
The manuscript intitled “Characterization and preliminary application of a novel lipoxygenase from Enterovibrio norvegicus” presents interesting results of a lipoxygenase characterization. These results are important because these enzymes can be applied to increase dough quality if the kinetic parameters are improved. They have shown that this novel lipoxygenase has increased thermostability and high activity. Although the results and conclusions are supported by data, a revision should be performed, specially related English writing. Some mistakes are comment here to show some examples, but many other should be checked.
Abstract
Species names should be in italics
Introduction
“Lipoxygenases (LOXs) are a family of non-heme iron dioxygenases, which can catalyze polyunsaturated fatty acids (PUFAs) with 1-cis, 4-cis-pentadiene structures into fatty acid hydroperoxides [1].” Which can catalyze...I think the type of reaction is missing in this sentence.
Sentence 29-32: very confusing. Please consider revision.
“Although researchers have made a lot of efforts to improve the characteristics of LOXs by protein engineering [13, 17-19], the LOXs have been reported could not meet the requirements of industrial application” Please, specify here the conditions required for industrial applications
“In this study, a LOX gene from E. norvegicus DSM 15893 (EnLOX) was cloned and expressed in E. coli BL21 (DE3).” It is not clear how this strategy would be good to improve the characteristics of LOXs.
Methodology
Line 73: “containing” instead of “contained”
“After cells were completely broken by high pressure cell cracker...” Specify which equipment was used for that.
Results
Table 1: Meaning of acronyms is missing
Figure 4: pH stability was measured for how long? Figure legend should be clear on that point. Also, stability parameters could be calculated for a better discussion.
Figure 4 and table 3: the activity related to 100% should be specified in legends.
Table 4: Meaning of acronyms is missing
Author Response
Thank you for your comments. We have been provided a point-by-point response to the reviewer’s comments and enter it in the box below.
Meanwhile, we upload point-by-point response as a PDF file, please see the attachment
Comments: The manuscript intitled “Characterization and preliminary application of a novel lipoxygenase from Enterovibrio norvegicus” presents interesting results of a lipoxygenase characterization. These results are important because these enzymes can be applied to increase dough quality if the kinetic parameters are improved. They have shown that this novel lipoxygenase has increased thermostability and high activity. Although the results and conclusions are supported by data, a revision should be performed, specially related English writing. Some mistakes are comment here to show some examples, but many other should be checked.
Response: We appreciate the referee’s enthusiasm for our work. Thanks for these great suggestions. Below are our detailed point-to-point responses. And we have invited native English speaker to revise the language of the manuscript.
Abstract
Point 1: Species names should be in italics
Response 1: Thanks for the referee’s careful correction. We apologize for these mistakes. We have revised them in the revised manuscript (page 4, line 11-12).
Line 11-12: Here, a lipoxygenase (LOX) gene from Enterovibrio norvegicus DSM 15893 (EnLOX) was cloned and expressed in Escherichia coli BL21 (DE3).
Introduction
Point 2: “Lipoxygenases (LOXs) are a family of non-heme iron dioxygenases, which can catalyze polyunsaturated fatty acids (PUFAs) with 1-cis, 4-cis-pentadiene structures into fatty acid hydroperoxides [1].” Which can catalyze...I think the type of reaction is missing in this sentence.
Response 2: Thanks for the referee’s comment. We're sorry that we didn't make it clear. In the revised manuscript, we have rewritten it into “Lipoxygenases (LOXs) are a family of non-heme iron dioxygenases, which catalyze the regiospecific and stereospecific insertion of molecular oxygen in polyunsaturated fatty acids (PUFAs) containing 1-Z, 4-Z-pentadiene moieties into hydroperoxides fatty acid with (Z), (E)-diene conjugations [1-3].” (p. 1, line 25-28)
Point 3: Sentence 29-32: very confusing. Please consider revision.
Response 3: Thanks for the referee’s suggestion. We have revised it in the manuscript (p. 1, line 30-33).
Line 30-33: “LOX is extracted mainly from soybean powder [6-9], which contains multiple enzymes, may reduce the catalyzing effects of LOX, and produce unpleasant odor after the catalytic process, thus limit the application of LOXs in flour roducts [7, 10, 11].”
Point 4: “Although researchers have made a lot of efforts to improve the characteristics of LOXs by protein engineering [13, 17-19], the LOXs have been reported could not meet the requirements of industrial application” Please, specify here the conditions required for industrial applications.
Response 4: Thanks for referee’s advice. We're sorry that we didn't make it clear. We have added this part in the revised text (p. 2, line 50-52). And the corresponding references have been also cited in the manuscript (p. 12, line 391-392; p. 12, line 421-422).
Line 50-52: Therefore, microbial-LOXs has been reported could not be suitable for industrial application, which calls for high yield, high purity, acid and alkali resistance, good thermal stability, and high catalytic activity [7, 21].
Line 391-392:
- Shi, K.; Wang, P.; Zhang, C.; Lu, Z.; Chen, M.; Lu, F., Effects of anabaena lipoxygenase on whole wheat dough properties and bread quality. Food Sci. Nutr. 2020, 8, (10), 5434-5442.
Line 421-422:
- Qian, H.; Zhang, C.; Lu, Z.; Xia, B.; Bie, X.; Zhao, H.; Lu, F.; Yang, G. Y., Consensus design for improved thermostability of lipoxygenase from Anabaena sp. PCC 7120. BMC Biotechnol. 2018, 18, (1), 57-63.
Point 5: “In this study, a LOX gene from E. norvegicus DSM 15893 (EnLOX) was cloned and expressed in E. coli BL21 (DE3).” It is not clear how this strategy would be good to improve the characteristics of LOXs.
Response 5: Thanks for the referee’s suggestion. To date, many LOXs from microorganisms have been expressed in E. coli successfully. These microbial-LOXs is far superior to LOXs from animals and plants in terms of catalytic activity, but their thermostability and pH-stability is unsatisfactory. Although researchers have made a lot of efforts to improve the properties of LOXs, there is still no LOX suitable for industrial application. Therefore, a novel LOX gene was explored to develop enzymes with commercial value.
The mining of novel enzyme generally involves the following steps:
(1) Firstly, the novel of the LOX was analyzed using bioinformatics technology.
(2) Secondly, the gene coding for the enzyme was cloned into a suitable expression vector by genetic engineering techniques and subsequently transformed into a suitable expression host cell for the overexpression. E. coli was chosen as the expression host to ensure the yield and purity of the LOX according to the reported references (LWT -Food Sci. Technol. 2019,109, 415-421; Int. J. of Biol. Macromol. 2020, 143, 685-695; Int. J. of Biol. Macromol. 2020, 156, 812-828).
(3) Finally, the LOX was purified and characterized.
In this study, to make it more clearly, we have revised it in the manuscript (p. 2, line 56-58).
Line 56-58: In this study, a novel LOX gene from E. norvegicus DSM 15893 was analyzed using bioinformatics technology, then cloned and expressed in E. coli BL21 (DE3). The properties of the recombinant E. norvegicus LOX (EnLOX) have been characterized.
Methodology
Point 6: Line 73: “containing” instead of “contained”
Response 6: Thanks for the referee’s advice. We have revised it (p. 3, line 85)
Point 7: “After cells were completely broken by high pressure cell cracker...” Specify which equipment was used for that.
Response 7: Thanks for the referee’s suggestion. We have added the name, model, and supplier of the equipment in the manuscript (p. 3, line 89-90).
Line 89-90: After cells were completely broken by D-3L High Pressure Homogenizer (PhD Technology LLC, USA), cell debris were removed by centrifugation (4°C, 10000×g, 30 min).
Results
Point 8: Table 1: Meaning of acronyms is missing
Response 8: Thanks for the referee’s advice. We have added the meaning of LA, ALA, GLA in the caption of Table 1 (p. 8, line 253).
Line 253: LA is linoleic acid, ALA is α-linolenic acid, and GLA is γ-linolenic acid.
Point 9: Figure 4: pH stability was measured for how long? Figure legend should be clear on that point. Also, stability parameters could be calculated for a better discussion.
Response 9: Thanks for the referee’s comment. We're sorry that we didn't make it clear. We have added the conditions of pH stability in the caption of Figure 5 (p. 9, line 297-298).
Residual activity was used as a stability parameter to further discuss the pH stability of EnLOX, and the corresponding description have been added in the manuscript (p. 8, line 283-286).
Line 297-298: Enzyme samples were incubated at different temperatures (20-70°C) for different times (0-12 h) before a 5 min ice bath, the residual activities were measured in 50 mmol L-1 Tris-HCl (pH 8.0) at 50°C.
Line 283-286: EnLOX displayed excellent stability in assayed pH range (4.0-9.5), it could retain more than 50% relative activity after being incubated at 4°C for 10 h, and it could retain above 70% relative activity at a broad pH range (5.0-9.5). Surprisingly, EnLOX remained above 90% relative activity at pH 7.5-8.0 after stored at 4°C for 10 h (Figure 5d).
Point 10: Figure 4 and table 3: the activity related to 100% should be specified in legends.
Response 10: Thanks for referee’s suggestion. We apologize that we did not make it clear. We have supplemented the relevant description in the legends of Figure 5 (p. 9, line 293-294) and Table 3 (p. 9, line 316-317).
Point 11: Table 4: Meaning of acronyms is missing
Response 11: Thanks for the referee’s advice. We have supplemented the meaning of WA, DDT, ST, SD in the caption Table 4 (p. 10, line 335-336).
Line 335-336: WA is water absorption, DDT is dough development time, ST is stability time, and SD is softening degree.

Reviewer 2 Report
The manuscript describes the characterization and preliminary application of a novel lipoxygenase from Enterovibrio norvegicus. The manuscript is interesting in the process viewpoint. After minor amendments, it could be processed further. Comments as follows:
1.Introduction - too short, explain more details about problem statement and current technology related to LOXs.
2. Provide flow chart of methodology, so that reader could understand better.
3. Line 93 - provide Michaelis-Menten equation and citation.
4. Provide citation for methodology used. i.e. Section 2.7, 2.9, 2.10
5. Why you choose Duncan's test rather than the other post-hoc test?
6. Figure 1 not clear.
7. Line 293 - provide reason why - Surprisingly, EnLOX remained above 50% relative activity at pH 4.0-9.5 263 after stored at 4°C for 10 h
8. Figure 5 not clear.
9. Overall, discussion is sufficient.
10. Conclusion - too short, provide 'real values' in conclusion, should answer objectives.
11. Check format for references - spacing?
Author Response
Thank you for your comments. We have been provided a point-by-point response to the reviewer’s comments, and upload it as a PDF file, please see the attachment.

Reviewer 3 Report
Very good study, just minor revision made in the text

Author Response
Thank you for your comments. We have been provided a point-by-point response to the reviewer’s comments, and enter it in the box below. Meanwhile, we upload it as a PDF file, please see the attachment.
Comments: Very good study, just minor revision made in the text.
Response: We appreciate the referee’s enthusiasm for our work. Thanks for these great suggestions. We revised the sentences which you marked. At the same time, we invited native English speaker to revise the language of the manuscript.
The “melting temperature” was replaced by “melting point temperature” in the revised manuscript (p. 1, line 15; p. 4, line 143, line 146; p. 8, line 275).
We have changed “for” into “due to” in the revised manuscript (p. 1, line 19).
We have added the word “monitoring” in the revised manuscript (p. 3, line 95).
The repetitive word “buffer” has been deleted in the revised manuscript (p. 4, line 128). Thank you again for your time.
We have changed “Increased” into “Increasing” in the revised manuscript (p. 5, line 212).
To better understanding the “the activity of EnLOX”, we have revised this sentence in the manuscript. (p. 8, line 280)
Line 280: The activity of EnLOX was assayed at different pHs, as shown in Figure 4b
